# Metformin Prevents NDEA-Induced Memory Impairments Associated with Attenuating Beta-Amyloid, Tumor Necrosis Factor-Alpha, and Interleukin-6 Levels in the Hippocampus of Rats

**DOI:** 10.3390/biom13091289

**Published:** 2023-08-24

**Authors:** Teresa Ponce-Lopez, José Antonio González Álvarez Tostado, Fernando Dias, Keren Happuck Montiel Maltez

**Affiliations:** Centro de Investigación en Ciencias de la Salud (CICSA), Facultad de Ciencias de la Salud, Universidad Anáhuac México Norte, Avenida Universidad Anáhuac 46, Lomas Anáhuac, Huixquilucan C.P. 52786, Estado de México, Mexico

**Keywords:** metformin, N-nitrosodiethylamine, memory impairment, β-amyloid, TNF-α, interleukin-6

## Abstract

N-nitrosodiethylamine (NDEA) is a potential carcinogen known to cause liver tumors and chronic inflammation, diabetes, cognitive problems, and signs like Alzheimer’s disease (AD) in animals. This compound is classified as probably carcinogenic to humans. Usual sources of exposure include food, beer, tobacco, personal care products, water, and medications. AD is characterized by cognitive decline, amyloid-β (Aβ) deposit, tau hyperphosphorylation, and cell loss. This is accompanied by neuroinflammation, which involves release of microglial cytokines, such as tumor necrosis factor-alpha (TNF-α), interleukin-6 (IL-6), and interleukin 1β (IL-1β), by nuclear factor kappa B (NF-κB) upregulation; each are linked to AD progression. Weak PI3K/Akt insulin-signaling inhibits IRS-1 phosphorylation, activates GSK3β and promotes tau hyperphosphorylation. Metformin, an antihyperglycemic agent, has potent anti-inflammatory efficacy. It reduces proinflammatory cytokines such as IL-6, IL-1β, and TNF-α via NF-κB inhibition. Metformin also reduces reactive oxidative species (ROS) and modulates cognitive disorders reported due to brain insulin resistance links. Our study examined how NDEA affects spatial memory in Wistar rats. We found that all NDEA doses tested impaired memory. The 80 µg/kg dose of NDEA increased levels of Aβ1-42, TNF-α, and IL-6 in the hippocampus, which correlated with memory loss. Nonetheless, treatment with 100 mg/kg of metformin attenuated the levels of pro-inflammatory cytokines and Aβ1-42, and enhanced memory. It suggests that metformin may protect against NDEA-triggered memory issues and brain inflammation.

## 1. Introduction

N-nitrosodiethylamine (NDEA) is an organic compound and a member of the N-nitrosamines, which refers to any molecule containing a nitroso functional group [1]. Nitrosamines are described as potential stomach and liver carcinogens and are classified as Group 2A carcinogens (probably carcinogenic to humans) by the International Agency for Research on Cancer (IARC) [2]. Their toxic and mutagenic effects occur through the alkylation of N-7 guanine, leading to destabilization and subsequent DNA breakdown [3]. The route of administration, dose, chemical nature of the compound, and frequency of exposure mediate the mutagenic effect of nitrosamines [1]. Exposure to nitrosamines can occur exogenously through ingestion; they have been extensively studied due to their presence in food and beer [4], tobacco [5,6], personal care products [7], water [8], and, more recently, in drugs such as angiotensin II receptor blockers (valsartan, losartan, and irbesartan), ranitidine, and metformin. The U.S. Food and Drug Administration confirmed the levels of NDEA and N-nitrosodimethylamine (NDMA) exceeding the interim acceptable ingestion limits; affected medicines were recalled [9,10]. Endogenously, they are synthesized from dietary precursors via chemical reactions between nitrites and secondary amines or proteins, resulting in the formation of endogenous carcinogenic N-nitroso compounds in a precise environment [11,12]. NDEA has the potential to produce reproducible liver neoplasms through the production of reactive oxygen species (ROS) and lipid peroxidation mechanisms [13,14]. A dose of NDEA of 200 mg/kg has been reported to induce liver tumors [14] and could incite chronic inflammation, elevating levels of interleukin-1β (IL-1β), interleukin-6 (IL-6), and tumor necrosis factor-α (TNF-α), and thereby attracting immune cells to the liver tissue [15]. Low doses (20 µg/kg) and diminished exposure to NDEA have been shown to cause type 2 diabetes mellitus (T2DM), cognitive impairments, and Alzheimer’s disease (AD)-type neurodegeneration with peripheral and cerebral insulin resistance in animal models [16,17].

AD is the most common neurodegenerative disease and is clinically manifested by a progressive deterioration in cognition and memory as well as progressive behavior changes [18]. One of the main neuropathological characteristics of AD is the accumulation of numerous forms of amyloid-β (Aβ) peptide in the brain, which can form amyloid plaques [19]. These plaques have toxic properties [20] that activate hyperphosphorylation of the tau protein, leading to the formation of neurofibrillary tangles (NFTs) [21]. Their accumulation in neurons leads to synaptic loss, interruptions to the conduction of impulses, and cell death [22]. Various signaling pathways become deregulated, impacting memory and triggering neurodegeneration [23]. Scientists have recognized insulin resistance as a significant factor contributing to cognitive problems in individuals with T2DM [24]. Impaired insulin signaling in the brain may lead to cognitive decline through various mechanisms, including reduced hippocampal neuroplasticity, increased tau protein levels, neuroinflammation, and mitochondrial dysfunction [25]. Accordingly, in PI3K/Akt brain insulin signaling, weak insulin signaling reduces insulin receptor substrate 1 (IRS-1) phosphorylation, resulting in the inhibition of the PI3K/Akt signaling pathway. Akt inhibition then activates GSK3β, which promotes tau hyperphosphorylation. It is also implicated in increased Aβ deposition and inflammation [26]. Another target that could be affected is mammalian rapamycin complex 1 (mTORC1), which regulates protein synthesis, synaptic plasticity, and autophagy [27]. A relationship between neuroinflammation and AD has been identified. Neuroinflammation is a process where brain-resident macrophages and microglia are activated to detect and remove harmful substances in the central nervous system [28]. The activation of the microglial production of proinflammatory cytokines accompanies the formation of Aβ plaques and NFTs. This neurotoxicity increases ROS production and the development of oxidative stress [29]. It is becoming increasingly clear that microglia-mediated neuroinflammatory cascades contribute to AD pathogenesis [30,31]. Several studies have linked the release of proinflammatory cytokines such as TNF-α, IL-1β, and IL-6 from microglia to the pathogenesis of AD [32]. Nuclear factor kappa B (NF-κB) is a well-established inflammatory transcription factor that can upregulate the expression of various proinflammatory genes, potentially contributing to neurodegenerative conditions [27].

Metformin belongs to the biguanide class of drugs. It is the first-line treatment for T2DM and is the most prescribed oral anti-diabetic medication worldwide. The key mechanism through which metformin reduces blood glucose levels is by inhibiting liver glucose production and augmenting glucose utilization [33,34]. The drug achieves its therapeutic effect primarily by activating AMP-activated protein kinase (AMPK). When activated in the liver, AMPK suppresses gluconeogenesis and fatty acid synthesis. In contrast, in skeletal muscle tissue, AMPK activation enhances glucose uptake by increasing the presence of glucose transporter 4 (GLUT4) in the cell membrane [35,36]. Metformin possesses several other remarkable properties, including its capacity to reduce ROS production [37] and regulate inflammatory pathways [38]. Numerous studies have demonstrated that metformin decreases the levels of proinflammatory cytokines such as IL-6, IL-1β, and TNF-α, both in preclinical [37,39] and clinical studies [40,41]. Metformin is believed to inhibit the activity of NF-κB. It achieves this by blocking the phosphorylation of the nuclear factor of the kappa light polypeptide gene enhancer in B cells (IκB) and IκB kinase (IKK)α/β [42,43]. This blockade impedes the migration of NF-κB into the nucleus, resulting in a reduced expression of proinflammatory genes [44]. Due to the association between brain insulin resistance and cognitive impairments, there is growing interest in repurposing existing anti-diabetic medication as potential treatments for cognitive-decline disorders [45,46,47].

In this study, we investigated the impact of exposure to different doses of NDEA on spatial memory, hippocampal Aβ levels, and proinflammatory cytokines such as TNF-α and IL-6 in rats. We examined the effect of metformin on memory impairments as well as Aβ1-42 and proinflammatory cytokines induced by NDEA.

## 2. Materials and Methods

### 2.1. Drugs

NDEA and metformin were obtained from Sigma Chemical Co. (St. Louis, MO, USA). NDEA (442 687) ampulla 1 mL (1000 mg/mL); NaCl0.9% (*w*/*v*) was used for dilutions and was stored in the dark. Metformin hydrochloride 200 mg (1115-70-4) was dissolved in NaCl0.9% (*w*/*v*) before use.

### 2.2. Animals

Male Wistar rats aged 12 weeks and weighing 180–200 g were used. The animals were obtained from the ADN S.A. de C.V animal house. The animals were housed in managed conditions with a room temperature of 22 2 °C, a 12 h light/dark cycle, and free access to food and water.

### 2.3. Experimental Design

The study consisted of two experiments. First, we explored the effect of different doses of NDEA on learning and spatial memory. The animals were randomized and divided into five groups (*n* = 8 in each group). Four groups received different injections of NDEA at doses of 20 µg/kg (group 1), 40 µg/kg (group 2), 60 µg/kg (group 3), and 80 µg/kg (group 4). These were intraperitoneal (i.p.) injections that were administered every 48 h on days 1, 3, and 5 (3 × x), following previously described methods [16,48]. The fifth group, performing as the control, received i.p. injections of the vehicle (3×). Second, we assessed whether NDEA induced an increase in Aβ1-42, TNF-α, and IL-6 as well as memory impairments, and whether metformin could reverse these effects. NDEA at a dose of 80 µg/kg was chosen to investigate the protective role of metformin as it exhibited impairments in restorative memory acquisition. The animals (*n* = 8 in each group) were randomly assigned to three groups as follows: (1) NDEA 80 μg/kg i.p. (3×); (2) NDEA 80 μg/kg i.p. (3×) plus metformin 100 mg/kg/day/orally (o.v.) for 14 days; and (3) a control group, which received an i.p. vehicle (3×) and an o.v. vehicle for 14 days. On day 10 of the metformin treatment, a memory test was initiated for all groups. The hippocampus was dissected from the euthanized rats on the fifth day of the behavioral test and stored at –80 °C for the ELISA assay (Figure 1).

### 2.4. Morris Water Maze Test

Memory performance was measured using a Morris water maze (MWM) [49]. It consisted of a circular pool with a diameter of approximately 120 cm and a height of 50 cm filled with water at a temperature of 23 ± 2 °C. The pool was divided into four equal quadrants; a submerged escape platform (10 cm in diameter) located 2 cm below the surface of the water was placed in the southwest (SW) quadrant. Three visual cues (a square, a circle, and a triangle) were positioned as reference points for the location of the platform. In this experiment, we used clear water. The platform was camouflaged using a transparent platform against the colored background; this was to prevent the animals from distinguishing the platform when swimming.

Training trials: The animals underwent four daily training trials for four consecutive days. Each trial involved placing the animals in the pool facing the pool wall in one of the selected quadrants (the starting position varied daily). The rats had a maximum of 60 s to locate the hidden platform and 20 s on the platform. The interval between the trials was 20 s. Rats unable to locate the platform within 60 s were manually guided to it and held there for 15 s. The time taken to reach the platform (escape latency) was measured as an indicator of spatial learning.

Probe trial: On the fifth day, the platform was removed. The animals were placed in the pool from the quadrant opposite the training quadrant and allowed to swim freely for 60 s. The time spent in the target (SW) quadrant was recorded as a measure of spatial memory. The trials were recorded and analyzed using SMART v3.0.02 Harvard apparatus software.

### 2.5. Brain Tissue Preparation 

On the 14th day, rats from the second experiment group were euthanized with pentobarbital (50 mg/kg i.p.) and decapitated. Their brains were quickly removed and the hippocampus was dissected. The tissue was frozen at −80 °C until further analysis. The tissue was then homogenized in 10 wet-weight volumes of tris buffer saline (pH 8.0) containing protease and phosphatase inhibitors. The resulting lysates were centrifuged and the supernatants were stored at −80 °C for the ELISA assay.

### 2.6. Enzyme-Linked Immunosorbent Assay (ELISA)

TNF-α and IL-6 were measured using a Rat ELISA kit from Invitrogen Co. (Waltham, MA, USA) (ab236712 and ab234570, respectively). The soluble Aβ1-42 levels were determined using the R&D Systems Quantikine^®^ ELISA (DAB142). The measured values were expressed as amounts per total protein. The assays were conducted as described in the instructions for use. Briefly, standards, controls, and hippocampal samples were added to the wells and incubated at 37 °C for 1.5 h. Biotin-detection antibody was added and incubated for 60 min at 37 °C. The plate was washed thrice using a wash buffer in each well and soaked for 1–2 min. The solution was added and incubated at 37 °C for 30 min. The plate was then washed five times as previously. TMB substrate was added, and the plate was incubated in the dark at 37 °C for 15–30 min. Finally, 50 μL of stop solution was added. The optical density of each well was read at 450 nm within 20 min.

### 2.7. Statistical Analysis

All results were analyzed using GraphPad Prism 8.0 software (San Diego, CA, USA). Data are expressed as the mean ± standard error of the mean (±SEM). A *p*-value of <0.05 was significant. A normal distribution was confirmed with the Shapiro–Wilk test. We conducted a two-way repeated-measure ANOVA followed by Bonferroni’s post hoc test to compare the acquisition data from the MWM test. A one-way ANOVA followed by Tukey’s post hoc test was used to analyze the percentage of time spent in the target quadrant in the MWM test as well as Aβ1-42, TNF-α, and IL-6 levels. The correlation between memory performance, Aβ1-42, and proinflammatory cytokines was assessed using Pearson’s correlation coefficient.

## 3. Results

### 3.1. NDEA Impairs Spatial Memory as Measured by the Morris Water Maze Test

The ability of animals to locate the hidden platform progressively decreased during the four days of the acquisition phase in the MWM, except the 20 µg/kg group. The differences in escape latency and traveled distance during training days 2–4 for all groups under treatments by NDEA were statistically significant compared with the control group, with [F(3, 21) = 13.31; *p* < 0.0001] for escape latency and [F(3, 21) = 14.42; *p* < 0.0001] for traveled distance. 

A repeated-measure two-way ANOVA followed by Bonferroni’s post hoc test for differences in escape latency demonstrated that compared to the control group, the performance in the NDEA 40 µg/kg and NDEA 60 µg/kg groups was significantly impaired [F (4, 28) = 8.535, *p* = 0.0001] on days 3 and 4, as it was in the NDEA 80 µg/kg group on days 2, 3, and 4 (Figure 2A; Table 1). Compared to the control group, the distance traveled was significantly different in the NDEA 40 µg/kg and NDEA 60 µg/kg groups [F(4, 28) = 9.249, *p* < 0.0001] on days 3 and 4, and in the NDEA 80 µg/kg group on days 2, 3 (Figure 2B; Table 1). The swimming speed did not change during the training trials as the training day progressed in the NDEA-treatment groups compared to the control group (Figure 2C), suggesting that the treatment did not produce any motor disruption in the animal groups [F(3, 21) = 2.089; *p* > 0.05].

Once the escape platform was removed, all NDEA groups spent significantly [F(3, 28) = 7.354, *p*= 0.0009] less time in the target quadrant, compared to the control group (43.8 ± 2.6 m). Times of the treatment groups were: NDEA 20 µg/kg, 35.9 ± 3 m; NDEA 40 µg/kg, 24.1 ± 4.3 m; NDEA 60 µg/kg, 20.2 ± 4.6 m; and NDEA 80 µg/kg, 18.2 ± 5.1 m. These results indicated significant impairment of learning and formation of memory reference with NDEA treatment (Figure 2D).

### 3.2. Metformin Treatment Rescues Spatial Memory in NDEA Rats

The effect of metformin (100 mg/kg) on cognitive deficits induced by NDEA (80 µg/kg) was tested using the MWM. The spatial learning skills of animals were assessed with four days of hidden platform tasks. A repeated-measure two-way ANOVA followed by a post hoc Tukey test demonstrated a change in the escape latency and the traveled distance during training days 2, 3, and 4 for the NDEA 80 µg/kg + metformin 100 mg/kg group. Between this group and the NDEA 80 µg/kg group, the difference in escape latency was statistically significant [F(1.425, 9.973) = 18.11; *p* = 0.0009], as it was in traveled distance [F(1.275, 8.920) = 22.43; *p* = 0.0007]. The NDEA 80 µg/kg group demonstrated a significantly longer escape latency and traveled distance on days 2 and 3, compared to the control group (Figure 3A,B; Table 2). Hence, treatment with metformin significantly reversed memory impairments in escape latency and traveled distance on days 3 and 4 compared with the control group in the hidden platform test (Figure 3A,B; Table 2).The analysis of swimming speed across the trials using a two-way ANOVA also demonstrated no significant difference among the groups as the training days advanced [F(1.233, 8.632) = 4.261; *p* > 0.05] (Figure 3C).

On the fifth day, the final day of the experiment, the hidden platform was removed to evaluate the memory of the trained rats. Figure 3D demonstrates that the NDEA 80 µg/kg group spent less time searching for the platform in the target quadrant (11.2 ± 3.8 s vs. 39.62 ± 3.7 s; *p* = 0.0035) than the control group. The NDEA 80 µg/kg + metformin 100 mg/kg group remained longer in the target quadrant (28.6 ± 2.1 s vs. 11.2 ± 3.8 s; *p* < 0.001) than the NDEA 80 µg/kg group (F(2, 21) = 1874; *p* < 0.0001). Treatment with metformin led to a partial recovery (*p* < 0.05) in the NDEA 80 µg/kg group compared to the control.

### 3.3. Metformin Reduces Brain Aβ1-42 Levels in NDEA Rats

The ELISA results revealed that soluble Aβ1-42 in the NDEA 80 µg/kg group was significantly higher (*p* < 0.0001; 103%) than in the control group. Treatment with metformin significantly restored Aβ1-42 levels (*p* < 0.05; 23%) in the hippocampus in comparison to the NDEA 80 µg/kg group (*p* < 0.001) and the control group (Figure 4A).

### 3.4. Metformin Reduces Proinflammatory Cytokine Levels

The data demonstrated that the levels of TNF-α were remarkably increased in the NDEA 80 µg/kg group by 83%, compared to the control group (*p* < 0.0001). Metformin significantly prevented a NDEA-induced rise in TNF-α in the hippocampus (*p* < 0.0001; 59%) (Figure 4B). The levels of IL-6 in the NDEA-treatment animals were significantly increased (*p* < 0.01; 30%) compared to the control group. Metformin significantly decreased IL-6 levels in the hippocampus of NDEA-treated rats (*p* < 0.01; 21%) (Figure 4C).

### 3.5. Correlations between Aβ, Inflammatory Markers, and Memory Consolidation

We examined the correlations between Aβ1-42 and proinflammatory cytokine levels and the time spent in the target quadrant. We observed significant inverse correlations between the Aβ1-42 (r = −0.7744; *p* < 0.01), TNF-α (r = −0.6876; *p* < 0.01), and IL-6 (r = −0.7578; *p* < 0.01) levels and the consolidation of memory in rats treated with NDEA 80 µg/kg group (Figure 5A–C). Only TNF-α demonstrated a significant correlation that was directly proportional to Aβ levels (r = 0.9359; *p* < 0.0001). IL-6 did not show a relationship with Aβ1-42 (r = 0.1687; *p* > 0.5) (Figure 6A,B).

## 4. Discussion

In this study, we investigated whether NDEA treatment (20, 40, 60, and 80 μg/kg i.p. x 3 (3×)) induced spatial memory deficits in young Wistar rats. We also evaluated the effect of NDEA (80 μg/kg) on hippocampal Aβ and proinflammatory cytokines and the ability of metformin to reverse these effects. The MWM test was used to assess the memory function. This is a valuable tool to evaluate spatial learning and memory [50], which relies on the integrity of the hippocampus and associated regions as well as prefrontal cortex connections [51]. As AD progresses, the hippocampal volume decreases, leading to amnestic syndrome [52]. It is susceptible to neurofibrillary tangles and Aβ deposition [53], which may induce neurotoxicity and can result in increased microglial activity linked to hippocampal atrophy and neuroinflammation in AD [54].

Our findings confirmed that exposure to low doses of NDEA affected the ability of the rats to learn and remember. NDEA at doses of 40, 60, and 80 μg/kg did not dependently affect memory consolidation. The chronic administration of metformin prevented reference memory disfunction in NDEA-treated 80 μg/kg group. Unlike hepatotoxic doses of NDEA (100–200 mg/kg), sub-mutagenic doses of NDEA have been linked to the development of insulin resistance in the brain. Tong et al. demonstrated that NDEA 20 μg/kg i.p. (3×)) caused deficits in motor functions and spatial learning 2–4 weeks later in Long Evans rat pups. Cerebellar and/or temporal lobe neurodegeneration was observed, including neuronal loss, oxidative stress, increased levels of phospho-tau and amyloid precursor protein–amyloid-β peptide (AβPP-Aβ), and choline acetyltransferase (ChAT) reduction [16]. Another study of NDEA treatment (15–250 μg/kg) effects on post-mitotic CNS neurons (48 h) cultured from Long Evans rat pups demonstrated dose-dependent impairments in ATP production and mitochondrial functions as well as increased levels of phospho-tau and AβPP-Aβ [17]. These effects were associated with a decreased expression of insulin, insulin-like growth factor (IGF) signaling, and ChAT [16,17].

NDEA 80 µg/kg caused a marked increase of Aβ1-42 and TNF-α levels and a moderate rise in IL-6 levels in the hippocampus. Metformin treatment reduced TNF-α levels more than IL-6 and Aβ1-42 levels. This suggests a potentially favorable impact of metformin on TNF-α modulation. Impairment of learning and formation of memory reference were associated with increased Aβ1-42, TNF-α, and IL-6 levels. Therefore, these markers may be involved in the detrimental effect on cognitive function. Furthermore, elevated TNF-α levels correlated with higher Aβ1-42 levels, supporting the hypothesis that inflammation can exacerbate amyloid pathology, leading to cognitive deficits. [55]. 

In the context of neurodegenerative conditions, neuroinflammation typically features the upregulation of proinflammatory cytokines such as IL-1, IL-6, TNF-α, interleukin-8 (IL-8), and transforming growth factor-beta (TGF-β). This is accompanied by the activation of microglia and astrocytes around Aβ plaques [32,56,57]. Aβ deposits in the brain have been linked to the activation of microglia. The binding of Aβ to the microglial cell surface induces the expression of proinflammatory genes, heightening the levels of proinflammatory cytokines such as TNF-α, IL-1β, IL-6, and IL-18. These actions result in tau hyperphosphorylation and neuronal loss [58]. It is now widely accepted that this process is a critical factor in the progression of AD, serving as a notable pathological marker [59]. The abnormal accumulation of Aβ and neurofibrillary tangles in the brain are the main triggers for neuroinflammatory responses in AD, activating resident glial cells [60]. This process is recognized as a contributing factor to neurodegeneration and compromises memory function in AD [30,31]. Thus, it is a rational proposition that NDEA (80 μg/kg) may trigger the release of hippocampal inflammatory markers from microglial cells, leading to memory impairments. 

Furthermore, our results were consistent with the neuroprotector and anti-inflammatory effects of metformin observed in numerous other studies. In a mouse model of Parkinson’s disease, metformin (150 mg/kg) reduced the amount of microglia, proinflammatory cytokines (TNF-α, IL-1β, and IL-6), and inducible nitric oxide synthase (iNOS) [61]. In AD APP/PS1 mice, metformin (200 mg/kg i.p.; 14 days) reduced Aβ levels and inflammatory responses (IL-1β and TNF-α), improving spatial memory and neurogenesis of the hippocampus. It was associated with increased levels of AMPK and reduced mTOR, NF-κB, and beta-secretase enzyme (Bace-1), which initiates the formation of Aβ oligomers [62,63]. Metformin acted as a regulator for the activation of IL-1β in the hippocampal cells of diabetic animals through a mechanism dependent on NF-κB [64]. Several mechanisms of action have been proposed for the anti-inflammatory effects of metformin. Metformin penetrates cells through organic cation transporters. Once inside, it acts on the mitochondria by blocking the respiratory chain complex 1. This action raises the internal ratio of AMP to ATP. Consequently, AMP kinase is activated, triggering various anti-inflammatory responses. Key among these responses is the suppression of the TNF-α/NF-κB and mTOR signaling pathways [65]. Another is via the activation of the mTOR pathway through AMPK-independent pathways, including the inhibition of transcription factor p65NF-κB [66] and PI3K/AKT of the insulin signaling pathway [66]. Hence, the justification for using metformin is its potential to slow aging processes through its role in mitochondrial metabolism, insulin signaling [45,67], and anti-inflammatory properties [64]. 

A significant limitation of our study was the inability to provide a more specific molecular data presentation of the signaling pathways mentioned above. Despite our best efforts, this limitation resulted in an unavoidable absence of molecular results. This omission limited our ability to obtain detailed conclusions about molecular interactions within our study data. To mitigate this, we attempted to describe broader findings that emerged from the molecular data in a simplified and accessible manner. 

## 5. Conclusions

Our research findings suggested that learning and memory impairment induced by NDEA was associated with high levels of Aβ1-42, TNF-α, and IL-6 in the hippocampus, and these effects were recovered by metformin. Hence, metformin could protect against NDEA-induced brain damage, indicating its potential as a neuroprotective and anti-inflammatory treatment in AD. Based on the existing literature, our results can be explained as a demonstration that metformin could improve memory dysfunction by mitigating proinflammatory cytokines production through the inactivation of NF-κB. This action could potentially enhance the insulin signal sensitivity via the PI3K/AKT/mTOR/Bace-1 pathway and suppress the production of Aβ1-42. Future research will focus on the NF-kB signaling pathways regulating TNF-α and Aβ.

## Figures and Tables

**Figure 1 biomolecules-13-01289-f001:**
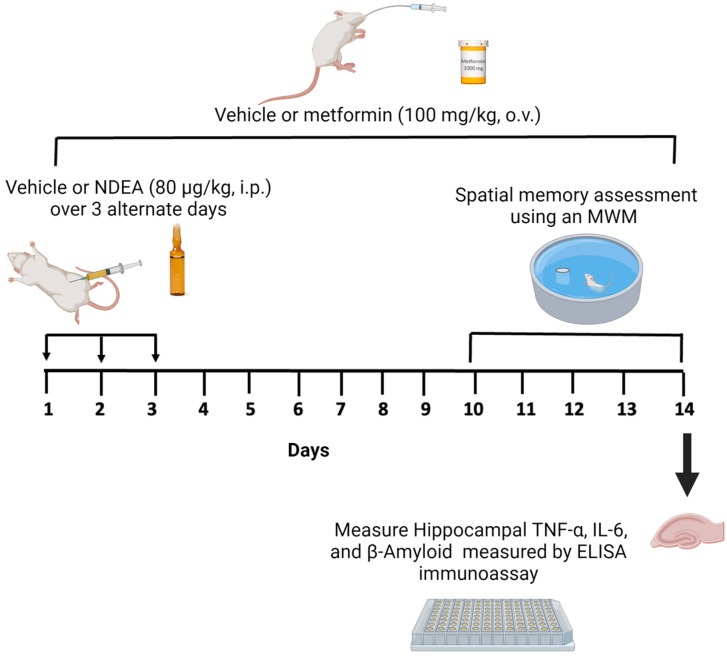
Schematic diagram of NDEA exposure, metformin treatment, and measurement of memory and markers of inflammation.

**Figure 2 biomolecules-13-01289-f002:**
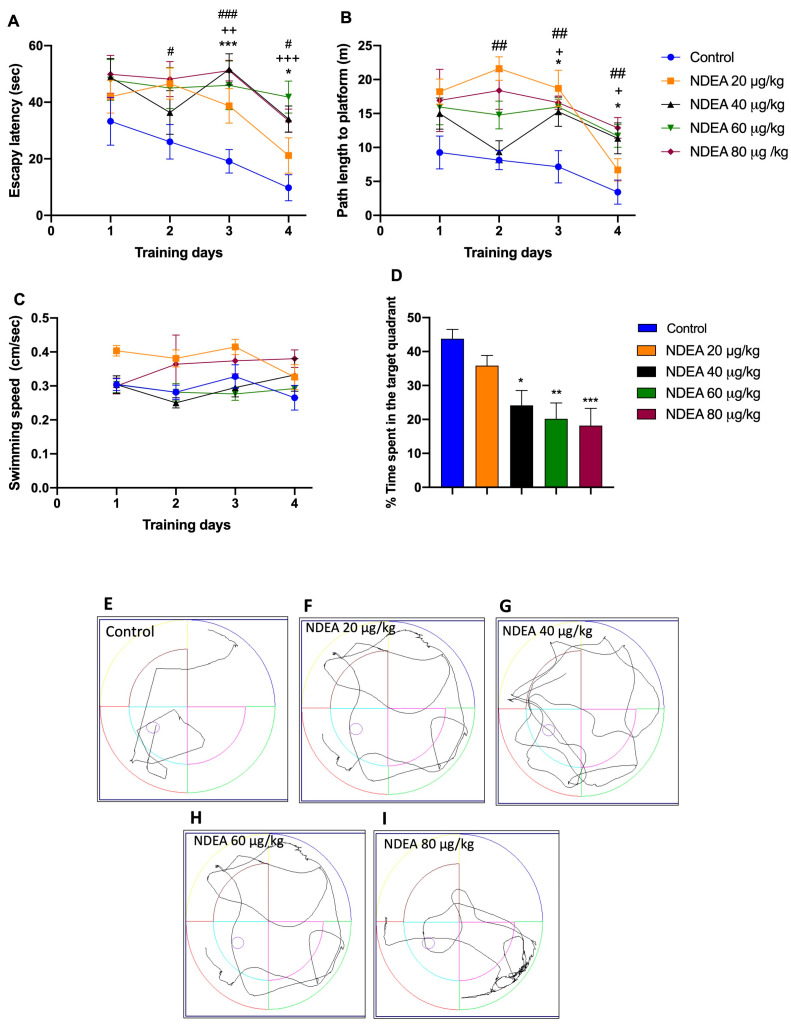
Effect of different NDEA doses (20, 40, 60, and 80 μg/kg) on performance in a reference memory task using an MWM. The animals received three intraperitoneal (i.p.) injections of either a vehicle or NDEA on alternate days. (**A**) Escape latency, (**B**) distance traveled, and (**C**) swimming speed of rats finding a hidden platform over four consecutive training days. The escape platform was in the southwest quadrant. The data represent the mean ± SEM (*n* = 8). Two-way ANOVA multiple comparisons followed by post hoc Bonferroni: NDEA 40 μg/kg * *p* < 0.05 and *** *p* < 0.001; NDEA 60 μg/kg + *p* < 0.05 ++ *p* < 0.01, and +++ *p* < 0.001; and NDEA 80 μg/kg # *p* < 0.05, ## *p* < 0.01 and ### *p* < 0.001 vs. control. (**D**) Percentage of time spent in the target platform quadrant for the fifth day. Tracing plots of the probe trials study groups: control (**E**), 20 μg/kg (**F**), 40 μg/kg (**G**), 60 μg/kg (**H**) and 80 μg/kg (**I**). One-way ANOVA followed by a post hoc Tukey test: * *p* < 0.05, ** *p* < 0.01 and *** *p* < 0.001 vs. control.

**Figure 3 biomolecules-13-01289-f003:**
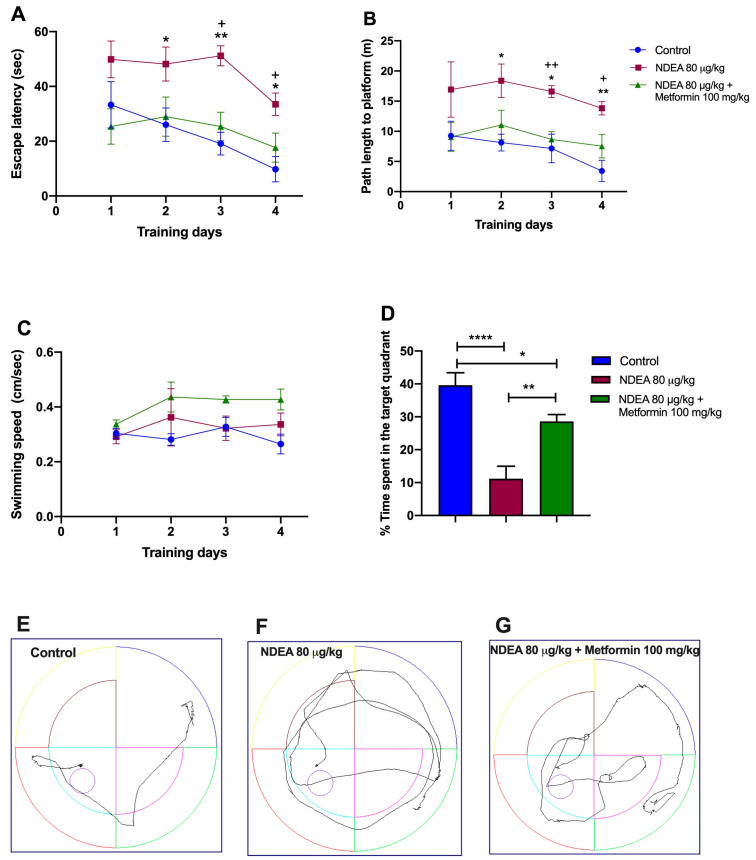
Effects of metformin on NDEA-injected rats. (**A**) Escape latency, (**B**) traveled distance, (**C**) swimming speed, (**D**) time in target quadrant, and tracing plots of the probe trials study groups: control (**E**), NDEA 80 μg/kg (**F**) and NDEA 80 μg/kg + metformin 100 mg/kg (**G**). The escape platform was in the southwest quadrant. The data represent the mean ± SEM (*n* = 8). Two-way ANOVA for repeated measures followed by post hoc Tukey test: * *p* < 0.05 and ** *p* < 0.01, NDEA 80 μg/kg vs. control; + *p* < 0.05 and ++ *p* < 0.01, NDEA 80 μg/kg vs. NDEA 80 μg/kg + metformin 100 mg/kg. One-way ANOVA followed by a post hoc Tukey test: **** *p* < 0.0001, NDEA 80 μg/kg vs. control; ** *p* < 0.01, NDEA 80 μg/kg vs. NDEA 80 μg/kg + metformin 100 mg/kg; * *p* < 0.05 NDEA 80 μg//kg + metformin 100 mg/kg vs. control.

**Figure 4 biomolecules-13-01289-f004:**
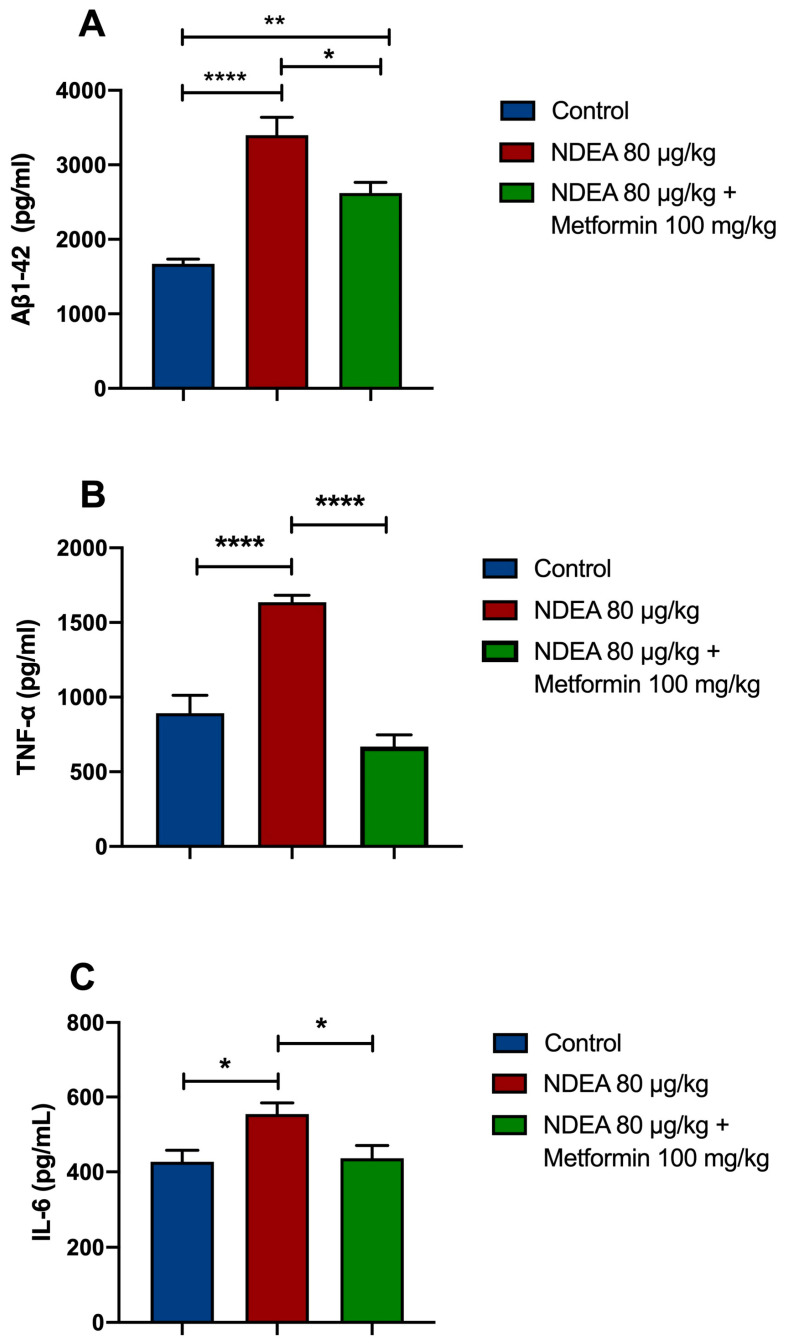
Effects of NDEA and metformin on (**A**) Aβ1-42, (**B**) TNF-α, and (**C**) IL-6 in the hippocampus of rats treated with three i.p. injections of 80 µg/kg NDEA. The data represent the mean ± SEM (*n* = 5–7). One-way ANOVA followed by a post hoc Tukey test: NDEA 80 µg/kg * *p* < 0.05, ** *p* < 0.01, and **** *p* < 0.0001 vs. control; ** *p* < 0.01 vs. NDEA 80 µg/kg vs. NDEA 80 µg/kg + metformin 100 mg.

**Figure 5 biomolecules-13-01289-f005:**
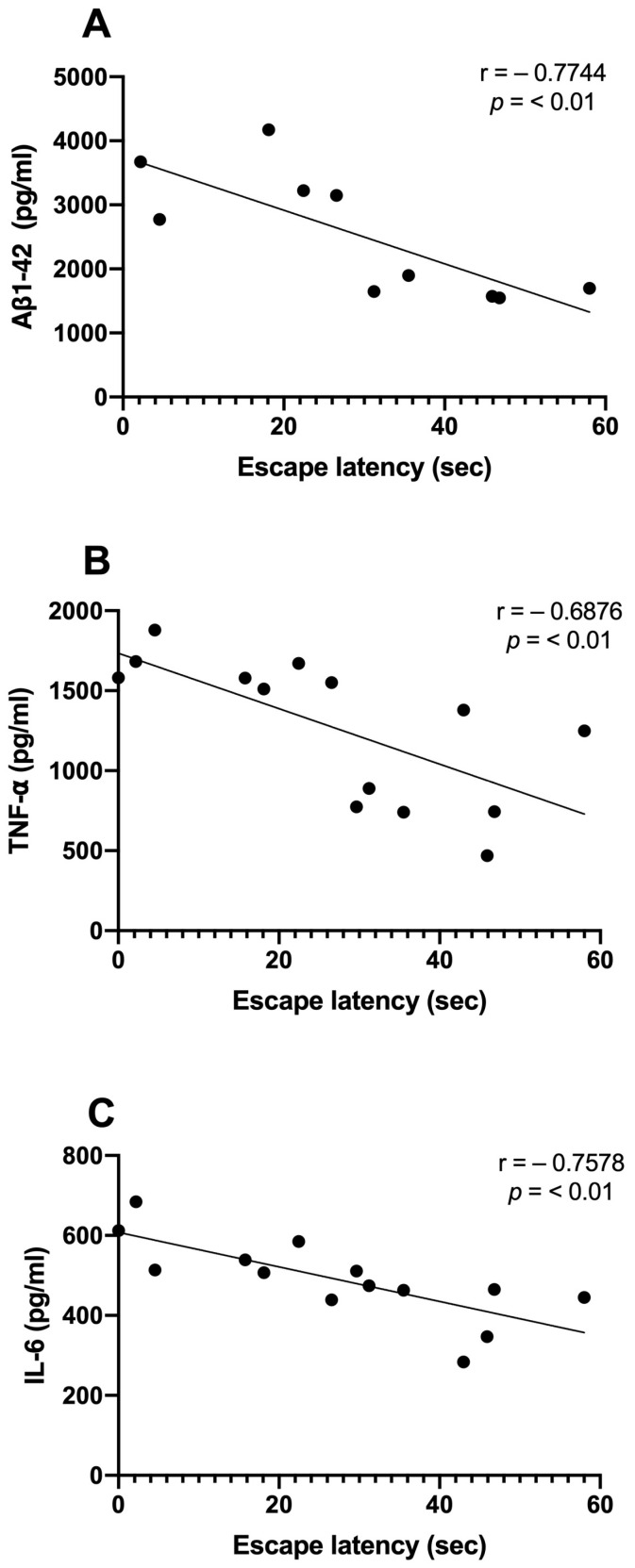
Relationship between hippocampal Aβ1-42, proinflammatory cytokine levels, and memory acquisition of NDEA-treated rats. Correlation analysis between levels of (**A**) amyloid fibril (Aβ1-42), *n* = 5; (**B**) tumor necrosis factor-alpha (TNF-α), *n* = 7; (**C**) interleukin-6 (IL-6), *n* = 7; and the percentage of time spent in the target platform quadrant on the last day of the test (r = −0. 7744; r = −0. 6976; r = −0. 7578; *p* < 0.01; Pearson’s correlation).

**Figure 6 biomolecules-13-01289-f006:**
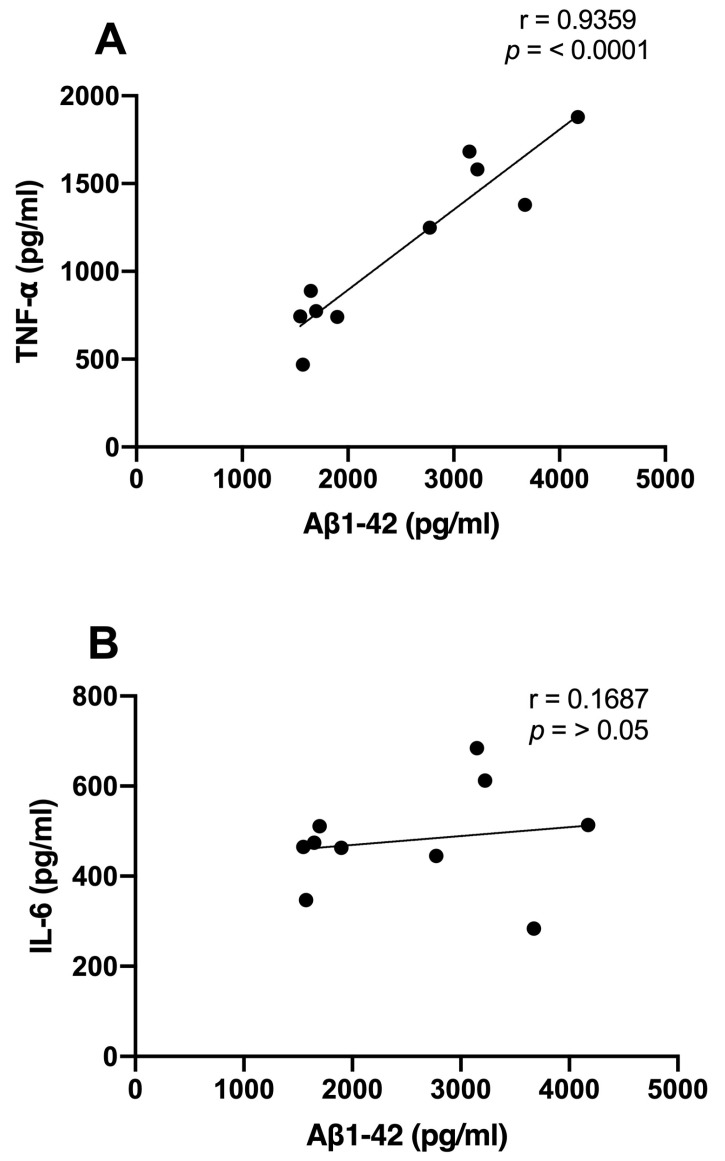
Relationship between hippocampal Aβ1-42 and proinflammatory cytokine levels of NDEA-treated rats. Correlation analysis between levels of (**A**) tumor necrosis factor-alpha (TNF-α), *n* = 7; (**B**) interleukin-6 (IL-6), *n* = 7; and amyloid fibril (Aβ1-42), *n* = 5 (r = 0. 9359; *p* < 0.0001; Pearson’s correlation).

**Table 1 biomolecules-13-01289-t001:** *p*-Values. Two-way ANOVA multiple comparisons followed by post hoc Bonferroni.

Training	Escape Latency		Traveled Distance	
**Day 1**	***p*-Values**		***p*-Values**	
Control vs. NDEA 20 μg/kg	ns	>0.9999	*	=0.0124
Control vs. NDEA 40 μg/kg	ns	=0.2291	ns	=0.1749
Control vs. NDEA 60 μg/kg	ns	=0.2955	ns	=0.0880
Control vs. NDEA 80 μg/kg	ns	=0.1726	*	=0.0396
**Day 2**				
Control vs. NDEA 20 μg/kg	ns	=0.0518	****	<0.0001
Control vs. NDEA 40 μg/kg	ns	=0.8223	ns	=0.982
Control vs. NDEA 60 μg/kg	ns	=0.0850	ns	=0.0904
Control vs. NDEA 80 μg/kg	*	=0.0303	**	=0.0033
**Day 3**				
Control vs. NDEA 20 μg/kg	ns	=0.0703	***	=0.0008
Control vs. NDEA 40 μg/kg	***	=0.0005	*	=0.0275
Control vs. NDEA 60 μg/kg	**	=0.0054	*	=0.0143
Control vs. NDEA 80 μg/kg	***	=0.0006	**	=0.0078
**Day 4**				
Control vs. NDEA 20 μg/kg	ns	=0.6468	ns	=0.6443
Control vs. NDEA 40 μg/kg	*	=0.014	*	=0.0322
Control vs. NDEA 60 μg/kg	****	=0.0006	*	=0.0237
Control vs. NDEA 80 μg/kg	*	=0.0172	**	=0.0076

ns: not significant; * *p* < 0.05; ** *p* < 0.01; *** *p* < 0.001; **** *p* < 0.0001.

**Table 2 biomolecules-13-01289-t002:** *p*-Values. Two-way ANOVA for repeated measures followed by post hoc Tukey test.

Training	Escape Latency		Traveled Distance	
**Day 1**	***p*-Values**		***p*-Values**	
Control vs. NDEA 80 μg/kg	ns	=0.4845	ns	=0.3419
Control vs. NDEA 80 μg/kg + Metformin 100 mg/kg	ns	=0.7811	ns	=0.9987
NDEA 80 μg/kg vs. NDEA 80 μg/kg + Metformin 100 mg/kg	*	=0.0384	ns	=0.4276
**Day 2**				
Control vs. NDEA 80 μg/kg	*	=0.0379	*	=0.0352
Control vs. NDEA 80 μg/kg + Metformin 100 mg/kg	ns	=0.9576	ns	=0.5966
NDEA 80 μg/kg vs. NDEA 80 μg/kg + Metformin mg/kg	ns	=0.1327	ns	=0.1438
**Day 3**				
Control vs. NDEA 80 μg/kg	**	=0.0083	*	=0.0128
Control vs. NDEA 80 μg/kg + Metformin 100 mg/kg	ns	=0.5468	ns	=0.8058
NDEA 80 μg/kg vs. NDEA 80 μg/kg + Metformin mg/kg	*	=0.0176	**	=0.0042
**Day 4**				
Control vs. NDEA 80 μg/kg	*	=0.0249	**	=0.0063
Control vs. NDEA 80 μg/kg + Metformin mg/kg	ns	=0.2969	*	=0.0207
NDEA 80 μg/kg vs. NDEA 80 μg/kg+ Metformin 100 mg/kg	*	=0.0274	*	=0.0483

ns: not significant; **p* < 0.05; ** *p* < 0.01.

## Data Availability

No new data were created or analyzed in this study. Data sharing is not applicable to this article.

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
