# Peer review of "Metformin Prevents NDEA-Induced Memory Impairments Associated with Attenuating Beta-Amyloid, Tumor Necrosis Factor-Alpha, and Interleukin-6 Levels in the Hippocampus of Rats"

_biomolecules, 2023, doi:10.3390/biom13091289_

Round 1

Reviewer 1 Report

Well planned, executed and presented work on a relevant topic. 

lines 207-209: ...NDEA 80 μg/kg group showed a significant increase in the escape latency on day 2 (p= 0.0379),  day 3 (p= 0.0083)  and  day 4 (p= 0.0249) and in the traveled distance on day 2 (p= 0.0352), day 3 (p= 0.0128) and day 4 (p= 0.0063), compared  to the control group (Figure 3A and B).

lines 211-213: escape latency on day 3 (p= 0.0176),  and day 4 (p= 0.0274) and in the traveled distance on day 3 (p= 0.0042) and day 4 (p= 0.0483), compared to the control group in the hidden platform  (Figure 3A and B).

229 line: please put the P value at the end of the sentence...remarkably increased in the NDEA 80 μg/kg group by 83%, compared to the control group (p<0.0001).

3.5 Please change title to: Correlations between Aβ, inflammatory markers and memory consolidation.

391-393 lines: please rephrase sentence.

399 line: please clarify " inflammatory nuclear pathway" 

Please make changes as described above. 

Reviewer 2 Report

biomolecules-2457663:Metformin prevents NDEA-induced memory impairment and neuroinflammation by reducing amyloid-β, tumor necrosis factor-α and interleukin-6 levels in rats´ hippocampus”

In this study, the authors are trying to demonstrate that metformin attenuates nneuroinflammatory effect of nitrosodiethylamine on the hippocampus, learning and memory. The material is described successively and conclusions are partially supported by obtained data.

Remarks/recommendations:

1.    in lines 24, (MCI) should be removed;

2.    in lines 94-97, the reference(s) should be added;

3.    in line 108, the source needs more details;

4.    in lines 117-118 and below, to avoid confusion the sentence(s) should be rewritten: “The animals were randomly assigned to five groups (n=8, in each)”.

5.    in line 134, the sentence needs a reference;

6.    In MWM test, opaque water is traditionally used to make invisible the escape platform (see J Neurosci Methods. 1984 11(1):47-60. doi: 10.1016/0165-0270(84)90007-4). In an alternative MWM version with clear water, the escape platform should be prepared from a transparent material. This should be clarified.

7.    in line 160, the source needs more details;

8.    in line 183 and below, ANOVA’s results should be described by F(x,y)= Z in addition to “p”;

9.    in lines 182-185 and below, the sentence should be rewritten (“p” values should be after “days”: “    on days 3 (p=0.0009) and 4 (p=0.0167)   “ and so on…;

10. in figures 2-4, the units of doses should be corrected (   /kg);

11. in figures 2-3, all plates next to “E” should be denoted by letters;

12. in lines 251and 270, the doses should be corrected;

13. in line 258, the sentence of “And tracing plots of the probe trial (E).” should be rewritten;

14. in line 296, “NDEA at doses of….”;

15. in lines 298-303, the sentence should be rewritten for clarity;

16. in lines 313-320 and 345-349, these “liver” fragments might be removed from the text;

17. in lines 364-380, this paragraph might be removed from the text;

18. in line 393, “(MCI)” should be removed while “EA” needs to be opened.

English should be double-checked.

Moderate editing of English language required

Reviewer 3 Report

This study presents the effect of Metformin on memory impairment and neuroinflammation through regulating the levels of Abeta, tumor necrosis factor alpha, and interleukin-6 in the presence of NDEA. The writing is straightforward and the experimental/theoretical approaches of the presented research make it attracted to the readership of the Biomolecules; however, major revision of the manuscript is necessary prior to publication.

Comments:

1.       The title should be revised. Based on the title, readers could think that Metformin could reduce Abeta, tumor necrosis factor alpha, and interleukin-6 levels by itself; however, the manuscript does not contain the experimental results. Please provide those information in the manuscript. Also, the authors should discuss that the effect of Metformin itself on the regulating the levels of Abeta, tumor necrosis factor alpha, and interleukin-6.

2.       Please provide the experimental results of Morris water maze test by treating Metformin to the rats without NDEA. That experiment could be considered as control experiment.

3.       In Figure 4 caption, the typo should be fixed.

4.       What is EA in the conclusions section?

5.       Some sentences should be revised. For example, the sentences start with “and” are not good ones.

English is well-written, however, please fix some typos and sentences.

Reviewer 4 Report

The manuscript from Ponce-Lopez and colleagues with the title: “Metformin prevents NDEA-induced memory impairment and neuroinflammation by reducing amyloid-β, tumor necrosis factor-α and interleukin-6 levels in rats´ hippocampus” addresses the effects of Metformin, a well-known anti-diabetic drug in preventing the cognitive decline and neuroinflammation induced in the rat hippocampus by low concentrations of N-Nitrosodiethylamine (NDEA). Their results confirm that the intraperitoneal administration of NDEA at doses ranging between 40 and 80 µg/Kg results in impaired hippocampus-dependent learning and memory associated with increased inflammation and Aβ levels in the hippocampus of rats. Furthermore, they show that co-administration of Metformin together with 80 µg / Kg prevents both the cognitive alterations and the brain pathology. All in all the experiments are well planned and seemingly carefully performed. The results are interesting, however not very novel. Indeed, an analysis of the mechanism of action of Metformin in this context would have added significantly to the interested of this manuscript.

Taken together, this manuscript presents a series of results supporting the interest in assessing the effects of neuroprotective and anti-inflammatory drugs in the treatment of cognitive impairments. I think that the manuscript should be accepted for publication after a few comments have been addressed.

Major comments

1)    In the methods a n=8 rats is given for both experiments. Am I correct in assuming that this means 8 rats per experimental group? Indeed a n of 8 rats divided into 5 or even 3 experimental groups would be unacceptable from the statistic point of view… Please make this clearer in the text. 

2)    Regarding figure 3D: did you test for significance between the control and the NDEA + metformin? Is it a full recovery? Or is the time spent in the target quadrant still significantly lower than in the controls?

3)    The entire manuscript, but especially the abstract and the result part need a thorough revision of the English writing. The abstract should also be re-written to better communicate the link between NDEA and the cognitive impairments/inflammation as well as between Metformin and inflammation. This come out quite nicely in the introduction, which is the part best written in the manuscript, but it is quite confusing in the abstract. All over the manuscript there are several phrases in which the grammatical structure of the phrase is not correct. This of course makes the reading particularly tiring.

Some examples:

a)    Line 166: the following phrase should be corrected “Data are expressed are presented…”

b)    Line 168: the following phrase should be corrected: “…we used a two-way repeated measure ANOVA followed by Bonferroni’s post hoc test was used to compare….”

c)    Line 176: the following phrase should be corrected: “All the NDEA animals’ group ability…”

d)    Line 193: the following phrase is not correct: “….., the comparison of all NDEA groups showed spent significantly….”

Etc….

4)     I would suggest to move the p values of the text and into a dedicated table. As they are now after each data point interrupt too much the flow of reading and complicate the understanding.

5)    Line 177: the data regarding the lowest dose of NDEA are given as “not shown”. I am not sure here what the policy of the journal is, but I think that all data should be showed. Maybe in a supplementary figure.

Minor points:

1)    Line 74: I am not sure to which “neurotoxicity” it is here referred

2)    Line 193: it is “escape” not “scape”

3)    Line 196: I think that “retardation of memory” is not the correct way of describing this result…the increased latency in finding the platform is a sign of impaired learning and formation of a reference memory.

4)    Line 210: “treated” should be “treatment”

5)    Line 300: “y” should be “and”

The grammatical structure of the phrases is very often not correct. The text should be thoroughly revised. In the present form the reading is extremely tiring and confusing.

Author Response

Please see the attachment." 

Round 2

Reviewer 2 Report

biomolecules-2457663: “Metformin prevents NDEA-induced memory impairment and neuroinflammation by reducing amyloid-β, tumor necrosis factor-α and interleukin-6 levels in rats´ hippocampus”.

The authors have made a careful revision and responded all points I raised.

Reviewer 3 Report

All my concerns were cleared. This revised manuscript is good to publish.